## Prospective observational study of the challenges in diagnosing common neonatal conditions in Nigeria and Kenya

Aimee P Staunton ![ORCID],[1] Helen M Nabwera ![ORCID],[2,3] Stephen J Allen,[2,4] Olukemi O Tongo,[5] Abimbola E Akindolire,[5] Isa Abdulkadir,[6] Chinyere V Ezeaka,[7] Beatrice N Ezenwa,[7] Iretiola B Fajolu,[7,8] Zainab O Imam,[9] Dominic D Umoru,[10] Walter Otieno,[11,12] Grace M Nalwa,[11,12] Macrine Olwala,[12] Alison W Talbert ![ORCID] ,[13] Pauline E A Andang'o,[14] Martha K Mwangome,[13] Ismaela Abubakar,[2] Nicholas D Embleton,[15,16] on behalf of the Neonatal Nutrition Network (NeoNuNet)

For numbered affiliations see end of article.

**Correspondence to**
Dr Aimee P Staunton;
aimee.staunton@ntlworld.com

## ABSTRACT

**Objectives** Accurate and timely diagnosis of common neonatal conditions is crucial for reducing neonatal deaths. In low/middle-income countries with limited resources, there is sparse information on how neonatal diagnoses are made. The aim of this study was to describe the diagnostic criteria used for common conditions in neonatal units (NNUs) in Nigeria and Kenya.

**Design** Prospective observational study. Standard case report forms for suspected sepsis, respiratory disorders, birth asphyxia and abdominal conditions were co-developed by the Neonatal Nutrition Network (https://www.lstmed.ac.uk/nnu) collaborators. Clinicians completed forms for all admissions to their NNUs. Key data were displayed using heatmaps.

**Setting** Five NNUs in Nigeria and two in Kenya comprising the Neonatal Nutrition Network.

**Participants** 2851 neonates, which included all neonates admitted to the seven NNUs over a 6-month period.

**Results** 1230 (43.1%) neonates had suspected sepsis, 874 (30.6%) respiratory conditions, 587 (20.6%) birth asphyxia and 71 (2.5%) abdominal conditions. For all conditions and across all NNUs, clinical criteria were used consistently with sparse use of laboratory and radiological criteria.

**Conclusion** Our findings highlight the reliance on clinical criteria and extremely limited use of diagnostic technologies for common conditions in NNUs in sub-Saharan Africa. This has implications for the management of neonatal conditions which often have overlapping clinical features. Strategies for implementation of diagnostic pathways and investment in affordable and sustainable diagnostics are needed to improve care for these vulnerable infants.

## STRENGTHS AND LIMITATIONS OF THIS STUDY

⇒ The use of heatmaps to display diagnostic pathways highlights key patterns in the diagnostic criteria used by clinicians.

⇒ Meticulous data collection, designed and supervised by local clinicians and international researchers, has provided fresh insights into diagnostic practice in this under researched population.

⇒ Most network neonatal units (NNUs) were urban, tertiary level facilities serving families that could likely better afford investigations so that our findings may not be generalisable to NNUs serving poorer populations.

## BACKGROUND

The majority of the 2.4 million global neonatal deaths (<28 days) in 2019 were due to three largely preventable or treatable conditions: complications of preterm birth, intrapartum-related events and sepsis.[1] Accurate and timely diagnosis of these common conditions is crucial for reducing mortality.[1 2] The clinical assessments and decisions from presentation to diagnosis, referred to as diagnostic pathways, are known to influence outcomes.[3] Although access to diagnostics is included in one of the six WHO health system building blocks, there are limited data on diagnostic pathways and how these influence outcomes in neonatal care in low/middle-income countries (LMICs).[4–6]

In Nigeria and Kenya, concerted actions are required to reduce neonatal mortality.[7] Progress in reducing neonatal mortality rates (NMRs; the number of deaths occurring 0–<28 days per 1000 live births) has been slow in both countries.[8] Between 1990 and 2019, NMR in Nigeria decreased by 28% from 50 to 36 per 1000 and in Kenya by 25% from 28 to 21 per 1000.[1] Consequently, both countries are unlikely to meet the sustainable development goal (SDG 3.2) for neonatal mortality

BMJ

and are thus classified as priority countries in the Countdown to 2030 initiative.[9]

In the context of robust health systems, national or regional guidelines may support the standardisation and accuracy of diagnoses as a means of improving care and reducing mortality in highly vulnerable hospitalised infants.[10 11] While both countries have policies on newborn health, only Kenya had national guidelines for the management of common neonatal conditions at the time of the study.[12 13] However, there is limited information on the impact of guidelines in low-resource settings, where challenges with resources for health are greatest.[14]

Encouraged by international efforts to improve access to diagnostics and the sparse information on how common neonatal diagnoses are made,[5 15–17] we sought to describe the diagnostic criteria used by clinicians in seven neonatal units (NNUs) in Nigeria and Kenya participating in the Neonatal Nutrition Network (NeoNuNet https://www.lstmed.ac.uk/nnu). Network members can be found in online supplemental appendix 1.

## METHODS
### Design and setting
In this prospective observational study, we collected information from five NNUs in Nigeria and two in Kenya; five providing tertiary level care, two providing district level care and all but one situated in major cities.[18] All NNUs provided supportive feeding, intravenous fluids, antibiotics, phototherapy and bubble continuous positive airway pressure. Two out of seven NNUs provide free neonatal healthcare services. Further details of the NNUs are available in online supplemental appendix 2.

### Data collection
Case report forms were developed for birth asphyxia, respiratory disorders, abdominal conditions and suspected sepsis with reference to international and local guidelines. Respiratory conditions included respiratory distress syndrome, transient tachypnoea of the newborn, pneumonia and meconium aspiration. Abdominal conditions included necrotising enterocolitis (NEC), dysmotility, septic ileus and focal intestinal perforation. Suspected sepsis was defined as clinical assessment resulting in starting or changing antibiotic treatment. Over a 6-month period in each NNU between September 2018 and April 2019, clinicians completed case report forms for all admissions and each episode of illness. A new episode occurred if the infant has been symptom free for 48 hours or more after any previous episode. Only the first episode of each illness was included in this analysis. Demographic and clinical information was extracted from the main NeoNuNet database. For out-born infants in whom birth weight was not known, weight on admission was used instead of birth weight. Completed case report forms were reviewed by a senior clinician. Data clerks entered anonymised data into a password protected REDCap database hosted by

the Liverpool School of Tropical Medicine.[10 19] Ethical approval was granted (online supplemental appendix 3).

### Patient and public involvement
Study participants and the public were not involved in the design or undertaking of the study.

### Data management and analysis
Data were imported to SPSS V.25[20] for analysis. Categorical variables were presented as frequencies and percentages. Diagnostic criteria were presented using heatmaps (Microsoft Office 16 Excel). Each criterion was classified as either used (with either a positive or negative finding) or not used. Criteria were grouped into clinical, laboratory, clinical/laboratory and radiological criteria. The 'COUNTIFS' function was used to calculate the proportion of cases in which the criterion was used, and a divergent colour scheme was applied to the heatmap cells.

## RESULTS
Over the 6 months' data collection period, 2851 neonates were admitted to the seven NNUs. The majority of newborns were boys (1626; 57.1%), most had been delivered in a health facility (2600; 91.2%) and just over half were vaginal unassisted deliveries (1586; 55.7%).

Overall, 1261 (45.3%) were preterm (<37 weeks' gestation), 1405 (49.3%) were low birth weight (<2.50 kg) and 473 (16.6%) infants died (table 1). The proportions of very preterm (VPT, <32 weeks' gestation) or very low birth weight (VLBW, <1500 g) infants varied considerably across the NNUs (table 2).

### Diagnostic criteria
Overall, forms were filled in for 1230 (43.1%) newborns diagnosed with suspected sepsis, 874 (30.6%) with respiratory conditions, 587 (20.6%) with birth asphyxia and 71 (2.5%) with abdominal conditions (table 2). The frequency of these diagnoses varied considerably between NNUs including those with similar proportions of VPT or VLBW infants. Overall, across all NNUs, diagnoses were based mainly on clinical criteria and a full blood count with limited use of other laboratory assays or radiological investigations (figure 1A–D).

For the diagnosis of suspected neonatal sepsis, clinical criteria were used uniformly in all NNUs (figure 1A). The exceptions were hypotension where use varied from 0% to 98% (online supplemental appendix 4A). Assessment of glucose intolerance, often using a point of care test, and parameters from a full blood count were the most used laboratory criteria. However, other laboratory parameters requiring biochemical analyses and microbiology were used much less frequently overall.

Similarly, for respiratory conditions, clinical criteria were used uniformly across NNUs (figure 1B; online supplemental appendix 4B). In marked contrast to the limited use of laboratory analyses, the use of chest

**Table 1** Newborn characteristics of 2851 admissions*

| Variable | |
| --- | --- |
| Male gender† | 1626 (57.1) |
| Place of delivery | |
| Health facility | 2600 (91.2) |
| Home | 165 (5.8) |
| Other | 86 (3.0) |
| Mode of delivery‡ | |
| Vaginal unassisted | 1586 (55.7) |
| Caesarean section | 1160 (40.7) |
| Vaginal assisted/instrumental | 101 (3.5) |
| Gestational age in weeks,§ median (IQR) | 37.0 (33.0–39.0) |
| Gestational age category§ | |
| Extreme preterm (<28 weeks) | 122 (4.3) |
| Very preterm (28–<32 weeks) | 378 (13.3) |
| Moderately preterm (32–<37 weeks) | 761 (26.7) |
| Birth weight or weight on admission in kg,¶ median (IQR) | 2.50 (1.67–31.7) |
| Birth weight or weight on admission category¶ (n, %) | |
| ELBW (<1000 g) | 116 (4.1) |
| VLBW (1000–1499 g) | 393 (13.8) |
| LBW (1500–2499 g) | 896 (31.5) |
| Antibiotics given after birth** | 1294 (45.4) |
| Final infant outcome†† | |
| Discharged with no morbidities | 2170 (76.1) |
| Discharged with morbidities | 89 (3.1) |
| Transferred out | 43 (1.5) |
| Absconded/discharged against medical advice | 55 (1.9) |
| Died | 473 (16.6) |

*Variables are shown as number (%) unless otherwise stated.
†Gender missing in 2.
‡Mode of delivery missing in 4.
§Gestational age missing for 68.
¶Birth weight and weight on admission missing in 3.
**Prophylactic antibiotics missing in 36.
††Final infant outcome missing in 21.
ELBW, Extremely low birth weight; LBW, Low birth weight; VLBW, very low birth weight.

X-ray appeared to be good overall but with some variability between NNUs.

The diagnosis of birth asphyxia was based on clinical evidence of encephalopathy and exclusion of other aetiologies. Evidence of multiorgan dysfunction was also used although this may have been based on the clinical components (acute kidney injury, respiratory distress, circulatory collapse and disseminated intravascular coagulation) rather than laboratory analyses. In contrast, use of cranial ultrasound scan varied widely ranging from 0% to 84% (figure 1C; online supplemental appendix 4C).

For the infants with an abdominal condition, of which about half were diagnosed as NEC (36/71, 50.7%), a range of clinical criteria were used in nearly all cases in most NNUs. Basic laboratory criteria derived from a full blood count were also frequently used but use of parameters of metabolic acidosis was more variable. Overall, radiology criteria were used less frequently (figure 1D, online supplemental appendix 4D).

## DISCUSSION

This study enabled us to explore diagnostic pathways and identify key challenges in the diagnosis of common neonatal conditions in the context of limited resources. Clinicians were generally constrained to a symptoms-based, clinical orientated diagnostic approach. This is consistent with the widely available WHO Pocket Book which encourages a syndromic approach and is a common source of diagnostic guidelines.[21] However, the Pocket Book is targeted at first level referral hospitals, whereas greater access to diagnostics is required for higher-level care. Furthermore, for some neonatal conditions such as respiratory distress syndrome, a clinical diagnosis may be appropriate; however, given the overlapping features between neonatal conditions and the potential lack of adequate clinical history for out-born patients in particular, laboratory and radiological investigations are important to support accurate diagnosis.

The consistent use of a wide range of clinical criteria across the NNUs was a positive finding suggesting uniformity in clinical practice. This is likely supported by use of national neonatal clinical guidelines although these were available in Kenya only. National paediatric diagnostic guidelines have been found to optimise practice in high-income settings; however, this may be more challenging in LMICs where diagnostic resources and skilled workforce remain inadequate making consistent adherence to clinical guidelines difficult.[22 23] This is further complicated by political instability, healthcare strikes and high staff turnover due to poor working conditions which may result in gaps in training on appropriate use of diagnostic technologies when they are available.[24 25]

In marked contrast with clinical criteria, the limited use of laboratory investigations beyond a full blood count shows that there were challenges implementing standardised diagnostic pathways. There was obvious overlap between conditions, so that investigations such as blood culture were used infrequently in both suspected sepsis and respiratory illness. Although some laboratory procedures were available in individual hospitals, in our experience, this is often short-term due to dependency on donor or research support. The limited use of laboratory investigations within the NNUs may be due to a lack of affordable and quality assured laboratory and specimen transport services, which are common barriers in LMICs.[5 26] This is likely to have implications on the accuracy of diagnoses, since a syndromic diagnostic approach has been linked to lower specificity, thereby resulting in more false positive cases.[27] The lack of laboratory diagnostic support in our study would likely result in the overdiagnosis of neonatal sepsis. A direct consequence is the overuse of antibiotics in this population, which could

**Table 2** Admission characteristics and frequency of common conditions according to neonatal unit*†

| Condition | Kenya | | Nigeria | | | | | Total |
|---|---|---|---|---|---|---|---|---|
| | 1 | 2 | 3 | 4 | 5 | 6 | 7 | |
| Total admissions | 291 | 613 | 181 | 732 | 384 | 238 | 412 | 2851 |
| Extreme or very preterm (<32 weeks' gestation) | 52 (17.9) | 94 (16.3) | 9 (5.1) | 124 (17.4) | 142 (37.1) | 69 (29.1) | 10 (2.5) | 500 (27.8)‡ |
| Extreme or very low birth/admission weight (<1500 g) | 60 (20.6) | 107 (17.5) | 13 (3.3) | 104 (14.2) | 128 (33.3) | 69 (29.0) | 28 (6.8) | 509 (28.5)§ |
| Suspected sepsis | 132 (45.4) | 421 (68.7) | 68 (37.6) | 405 (53.3) | 49 (12.8) | 46 (19.3) | 109 (26.5) | 1230 (43.1) |
| Respiratory conditions | 78 (26.8) | 364 (59.4) | 19 (10.5) | 210 (28.7) | 18 (4.7) | 162 (68.1) | 23 (5.6) | 874 (30.7) |
| Birth asphyxia | 48 (16.5) | 189 (30.8) | 20 (11.0) | 190 (26.0) | 41 (10.7) | 44 (18.5) | 55 (13.3) | 587 (20.6) |
| Abdominal conditions | 3 (1.0) | 4 (0.7) | 6 (3.3) | 32 (4.4) | 3 (0.8) | 14 (5.9) | 9 (2.2) | 71 (2.5) |

*Data are shown as number (%).
†More than one condition was recorded in many infants.
‡Gestation was missing for 35 infants in unit 2, 5 in unit 3, 20 in unit 4, 1 each in units 5 and 6 and 6 in unit 7.
§Birth/admission weight was missing for two infants in unit 2 and one infant in unit 7.

drive antimicrobial resistance.[28] Use of laboratory investigations to complement clinical assessment, as well as antibiotic stewardship programmes, would likely help.[29]

Respiratory conditions such as respiratory distress syndrome were typically diagnosed using clinical criteria but also supported by chest X-rays which are critically important in distinguishing the cause of respiratory distress, which includes a number of pathologies with overlapping clinical signs.[30] The diagnosis of birth asphyxia was also based mainly on clinical criteria and with limited use of cranial ultrasound. In addition to diagnosis, laboratory support with determining acid–base balance and blood electrolytes is required in the selection of infants for treatment approaches such as therapeutic hypothermia and monitoring treatment response.[31]

In contrast to the use of X-ray in respiratory conditions, abdominal X-ray, a key investigation for the diagnosis of conditions such as NEC, was only used consistently in one

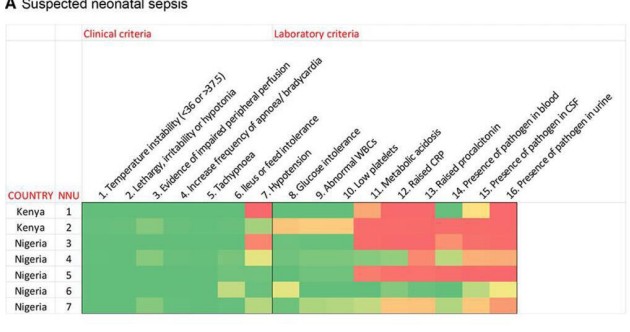

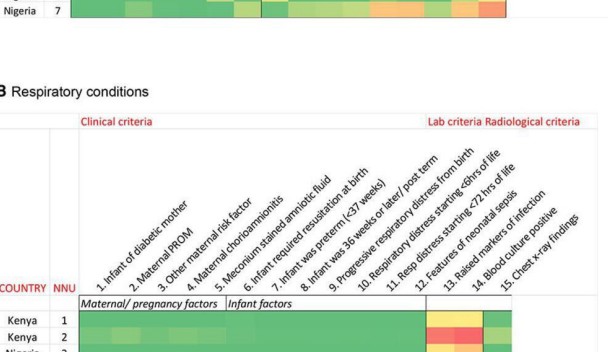

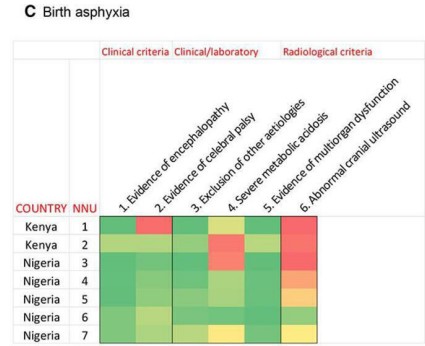

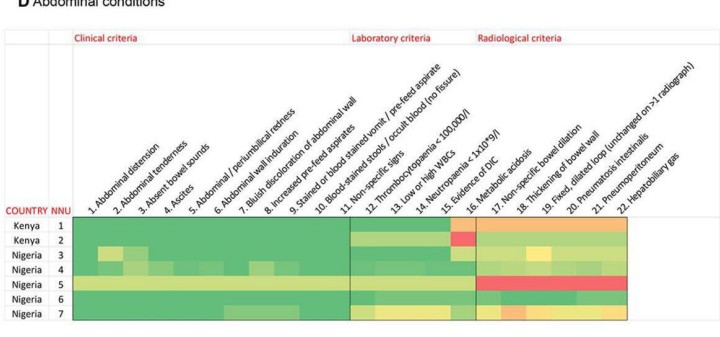

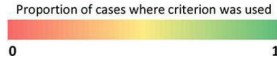

**Figure 1** Heatmaps showing the diagnostic criteria used for common neonatal conditions. (A) Suspected neonatal sepsis, (B) respiratory conditions, (C) birth asphyxia, (D) abdominal conditions.

unit. The limited use may be due to more severe illness in infants with suspected NEC and the lack of mobile X-ray machines. Previous research in LMIC settings has reported that diagnosis is impaired by a lack of abdominal X rays, since even when X-ray machines are available and functioning, they are often too expensive for patients and families to afford.[32] This is likely here since only two out of the seven NNUs provide free of charge neonatal health services. This means that diagnosis is very difficult for conditions such as NEC where specific X-ray findings are critically important.[33]

Given the consistent use of clinical criteria across the NNUs, the limited use of laboratory and radiological investigations likely contributed to the marked differences between units in the frequencies of diagnoses. This would appear to be less likely due to differences in case mix, since variability in the frequency of diagnoses occurred between NNUs with similar proportions of small and preterm newborns in whom these conditions are common. These challenges highlight the need for sustainable and affordable diagnostic tests suitable for use in resource limited settings for accurate and timely diagnosis and to reliably assess the frequency of the main causes of neonatal mortality. This should include point-of-care diagnostic tests designed for use in LMICs which can provide rapid results and reduce the workload for laboratory services.[34 35]

The limitations of our study relate mainly to the generalisability of our findings to other settings. Most of the network NNUs in this study were urban and tertiary level which are often inaccessible to disadvantaged families due to higher access costs. It is therefore likely that the families in our study could better afford some investigations than many in Nigeria and Kenya, limiting the generalisability of our findings. In addition, referral patterns, case mix and the experience of clinicians working within the NNUs in making diagnoses may also differ with other NNUs. Finally, although forms were completed by different cadres of clinical staff with varying levels of experience and senior review was limited to ensuring completeness of data, the use of the standardised study forms and review of data by senior clinicians may have encouraged the consistent use of clinical criteria in our study thereby improving clinical diagnoses compared with usual clinical practice.

## Conclusions

Our findings demonstrate that diagnostic pathways for common conditions in NNUs in Nigeria and Kenya were hindered by the limited use of laboratory and radiological technologies. Improving capacity to diagnose common conditions rapidly and reliably in hospitalised newborns should be a key priority of future healthcare quality improvement initiatives. The development of national plans to increase the availability, accessibility and affordability of point of care and other diagnostic investigations is paramount since the need for out-of-pocket health spending hinders compliance to clinical guidelines.[5]

This will be dependent on infrastructure strengthening and appropriate staff training and education. Our findings will inform the development and implementation of integrated context-relevant best practice guidelines for neonatal care in low-resource settings. Further research is necessary to examine the impact of such guidelines on neonatal health in these settings.

**Author affiliations**
[1]Department of International Public Health, Liverpool School of Tropical Medicine, Liverpool, UK
[2]Department of Clinical Sciences, Liverpool School of Tropical Medicine, Liverpool, UK
[3]Department of Infectious Diseases, Alder Hey Children's NHS Foundation Trust, Liverpool, UK
[4]Department of Gastroenterology, Alder Hey Children's NHS Foundation Trust, Liverpool, UK
[5]Institute of Child Health, University College Hospital Ibadan, Ibadan, Nigeria
[6]Department of Paediatrics, Ahmadu Bello University Teaching Hospital, Zaria, Nigeria
[7]Department of Paediatrics, Lagos University Teaching Hospital, Lagos, Nigeria
[8]Department of Paediatrics, College of Medicine University of Lagos, Lagos, Nigeria
[9]Department of Paediatrics, Lagos State University Teaching Hospital, Lagos, Nigeria
[10]Department of Paediatrics, Maitama District Hospital, Abuja, Nigeria
[11]Department of Paediatrics and Child Health, Maseno University, Maseno, Kenya
[12]Department of Paediatrics, Jaramogi Oginga Odinga Teaching and Referral Hospital, Kisumu, Kenya
[13]Department of Clinical Research, KEMRI-Wellcome Trust Research Programme, Kilifi, Kenya
[14]Department of Public Health, Maseno University, Maseno, Kenya
[15]Department of Paediatrics, Newcastle Upon Tyne Hospitals NHS Trust, Newcastle upon Tyne, UK
[16]Faculty of Medical Sciences, Newcastle University, Newcastle upon Tyne, UK

**Acknowledgements** We would like to thank our colleagues who contributed to the clinical care and collection of data at the network neonatal units. We would also like to thank colleagues at the Nigerian Society of Neonatal Medicine, the Kenya Paediatric Association and the Ministries of Health in Nigeria and Kenya, who provided us with support and advice as we were setting up this study. We thank the mothers and families immensely for participating in this study with their infants. This study is published with the permission of the Director of Kenya Medical Research Institute.

**Collaborators** Neonatal Nutrition Network Members: Isa Abdulkadir (Ahmadu Bello University, Nigeria); Ismaela Abubakar (LSTM, UK); Abimbola E Akindolire (College of Medicine, University of Ibadan, Nigeria); Olusegun Akinyinka (College of Medicine, University of Ibadan, Nigeria); Stephen J Allen (LSTM, UK); Pauline EA Andang'o (Maseno University, Kenya); Graham Devereux (LSTM, UK); Chinyere Ezeaka (Lagos University Teaching Hospital, Nigeria); Beatrice N Ezenwa (Lagos University Teaching Hospital, Nigeria); Iretiola B Fajolu (Lagos University Teaching Hospital, Nigeria); Zainab O Imam (Lagos State University Teaching Hospital, Nigeria); Kevin Mortimer (LSTM, UK); Martha K Mwangome (KEMRI Wellcome Trust Research Programme, Kenya); Helen M Nabwera (LSTM, UK); Grace M Nalwa (Jaramogi Oginga Odinga Teaching and Referral Hospital, Kenya & Maseno University, Kenya); Walter Otieno (Jaramogi Oginga Odinga Teaching and Referral Hospital, Kenya & Maseno University, Kenya); Alison W Talbert (KEMRI Wellcome Trust Research Programme, Kenya); Nicholas D Embleton (Newcastle University, UK); Olukemi O Tongo (College of Medicine, University of Ibadan, Nigeria); Dominic D Umoru (Maitama District Hospital, Nigeria); Janneke van de Wijgert (University of Liverpool, UK); Melissa Gladstone (University of Liverpool, UK).

**Contributors** SJA conceived the study. APS, SJA, HMN, OOT, AEA, IAbd, CVE, BNE, IBF, ZOI, DDU, WO, GMN, MO, AT, PEAA, MKM, IAbu and NE designed the study and each read, helped to critically revise and approved the final manuscript version and agree to be accountable for the accuracy and integrity of the manuscript and to answer any questions in relation to the work. APS analysed the data. APS, SJA and HMN prepared the manuscript. APS is the study guarantor and corresponding author responsible for submission, peer review and publication processes.

**Funding** This project was funded by a grant from the MRC Confidence in Global Nutrition and Health Research scheme (grant reference MC_PC_MR/R019789/1).

**Disclaimer** The funders had no role in the design or conduction of the study.

**Competing interests** None declared.

**Patient and public involvement** Patients and/or the public were not involved in the design, or conduct, or reporting, or dissemination plans of this research.

**Patient consent for publication** Not applicable.

**Ethics approval** This study involves human participants. Ethical approval was granted by the Research Ethics Committees at LSTM (18–0210), Lagos University Teaching Hospital Health REC (AMD/DCST/HREC/APP/2514), the Jaramogi Oginga Odinga Teaching and Referral Hospital (ERC.IB/VOL.1/510), University College Hospital Ibadan (UI/EC/18/0446), Massey Street Children's Hospital (LSHSC/2222/VOL.VIB/185), Ahmadu Bello University Teaching Hospital (ABUTH/HZ/HREC/D37/2018), Maitama District Hospital (FHREC/2018/01/108/19-09-18), the Kenya Medical Research Institute-Scientific and Ethics Review Unit (KEMRI/SERU/CGMR-C/120/3740). Participants gave informed consent to participate in the study before taking part.

**Provenance and peer review** Not commissioned; externally peer reviewed.

**Data availability statement** Data are available upon reasonable request. Further data from the heatmaps are available in appendix 4. Requests for access to further data should be addressed to the corresponding author.

**ORCID iDs**
Aimee P Staunton http://orcid.org/0000-0002-9782-0798
Helen M Nabwera http://orcid.org/0000-0003-1056-729X
Alison W Talbert http://orcid.org/0000-0002-9328-6903

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
