## [Reviewer comments · BMJ Open]

ARTICLE DETAILS

TITLE (PROVISIONAL)	A prospective observational study of the challenges in diagnosing common neonatal conditions in Nigeria and Kenya
AUTHORS	Staunton, Aimee; Nabwera, Helen; Allen, Stephen; Tongo, Olukemi; Akindolire, Abimbola; Abdulkadir, Isa; Ezeaka, Chinyere; Ezenwa, Beatrice; Fajolu, Iretiola; Imam, Zainab; Umoru, Dominic; Otieno, Walter; Nalwa, Grace; Olwala, Macrine; Talbert, Alison; Andang'o, Pauline; Mwangome, MK; Abubakar, Ismaela; Embleton, Nicholas

VERSION 1 – REVIEW

REVIEWER	Thomas, Deena Muthoot Healthcare, Neonatology
REVIEW RETURNED	06-Jul-2022

GENERAL COMMENTS	I congratulate the entire team of neonatal nutrition network for the conduct of such a meticulous study and the excellent presentation of the same in this study. However some minor suggestions from my side are: 1. The exact duration of study (from the date of enrollment of first neonate to the date of enrollment of last neonate) needs to be mentioned so as to avoid confusion between 6 or 8 months.2. In the design and setting highlight the retrospective nature of the study.3. Table 2 needs to mention in footnotes that the data of four conditions mentions the number of episodes and not the number of neonates because one neonate may have more than one condition or the same condition at different time points during the hospital stay (more than one episode of sepsis). Also data of the number of neonates admitted with any morbidity and the number of neonates free of any morbidity during hospital stay, needs to be highlighted in this table.4. Rephrase first line of the paragraph on diagnostic criteria. Reading the 'forms were completed' statement gives the impression that for the remaining neonates forms were incomplete.5. Appendix 3 needs to have footnote describing what 1 stands for.6. A bit of clarity needs to be brought in the data in heat maps in results section. In respiratory conditions in unit 1 in Kenya shows green in unit one for infant/maternal and pregnancy factors. However not all the cases of respiratory distress will have all the risk factors. Probably these are the criteria which were looked for in each of the following conditions by marking Yes or No as reflected by your case record forms and same should be highlighted in the manuscript.7. This statement "A new episode occurs if the infant has been symptom free for 48 hours or more after any previous episode" which is mentioned in your case report forms also need to be mentioned in methodology.
---

	8. Check the data for final outcome in Table 1 (it adds up to 2852). The study has implication in formation of countries' policy for improving diagnostic services and I once again congratulate the authors for the meticulous study.
--	--

REVIEWER	Tewari, Vishal Command Hospital (SC) and Armed Forces Medical College, Pediatrics
REVIEW RETURNED	11-Jul-2022

GENERAL COMMENTS	Reviewer comments Title of the manuscript: Challenges in diagnosing common neonatal conditions in Nigeria and Kenya The authors have conducted a study on a very relevant topic and have presented the study in a very lucid manner for which I congratulate them. However, several questions, comments are submitted for being addressed. Background  1. The study aim mention's ..'to describe the diagnostic criteria used by clinicians..'. What is the study hypothesis? 2. Author states...' However, there is limited information on the impact of guidelines in low-resource settings, where challenges with resources for health are greatest (14)'. How is this related to the current study? 3. The diagnosis of neonatal illness in inborn neonates' e.g. preterm respiratory distress syndrome is entirely clinical and hematological and radiological investigations support the diagnosis. In outborn neonates the role of lab investigations to establish and confirm the diagnosis is much more. 4. Patterns of neonatal morbidities in inborn and outborn neonates differ and while it is possible that without adequate lab, radiological, ultrasound and echocardiogram investigations establishing an accurate diagnosis may not be possible in outborn babies, diagnosis in inborn babies and initiation of treatment is almost always clinical. 5. Availability of guidelines for management of common neonatal conditions eventhough essential for improving neonatal care, what impact does its non-availability in a country in these times when open-source information and guidelines from WHO and similar nations are replete needs to be elaborated. 6. What international initiatives for improving diagnosis are being referred to? Material and methods  1. Design and setting: The study is a prospective observational study over a 6 month period from 5 tertiary care hospitals and 2 district hospitals from 2 African nations experiencing high NMR. Since the study is from Urban centers and tertiary care centers, the results of the study may have very little import on development of national policies for reducing NMR. This must be mentioned as a limitation of the study. 2. Design and setting: Did the tertiary care neonatal set-ups not have facilities for assisted invasive ventilation? 3. More details about the neonatal units enrolling the cases is required. How many bedded? What was the occupancy? Busy neonatal units may not be able to conduct all requisite hematological, biochemistry and radiographic examinations
---

	necessary for making a diagnosis. 4. In what proportion of cases was the diagnosis revised? 5. What was the reason for death and did the cause of death differ from the diagnosis at admission? 6. Was post-mortem examination done in the fatal cases? Was the provisional diagnosis revised following the post-mortem examination? 7. Investigations – hematological, biochemistry, acute phase reactants, radiological, ultrasound and echo play a vital role in guiding management but only help to complement the diagnosis in neonatal care algorithms. 8. Clinical staff completed the case reporting forms. Did it include Doctors and Nurses? What steps were taken to ensure accuracy of data entry in the forms? 9. Recording ‘.....a single episode for each condition in each infant...’ this sentence is not clear. Does it mean only one of the multiple diagnosis in a neonate was recoded? This can be a reason for reporting bias. 10. Diagnostic criteria used requires to be reanalyzed with:-  Distinction between Suspect, Possible and Confirmed sepsis, Distinction between neonatal encephalopathy of HIE vs. non-HIE origin. Distinction between Feed Intolerance, NEC and septic ileus, Distinction between preterm RDS, congenital pneumonia, TTN, MAS etc. 11. What parameters were used on cranial ultrasound for making the diagnosis of birth asphyxia? 12. What about mortality from neonatal lesions like congenital cardiac lesions, PPHN, congenital anomalies which could have any one or more of the clinical syndromes included in the study i.e. sepsis, respiratory distress, birth asphyxia and abdominal distention. Results 1. With more than 90% births in institutional setting and not home-births in this study, any further reduction in neonatal mortality rate will result from improvement in infrastructure and resources i.e. manpower, training and equipment. 2. Overall 16.6% mortality in institutional setting is exceptionally high and typical of resource constrained settings. 3. There is a remarkable difference in the morbidity patterns handled between the enrolling centers e.g. ELBW rates vary from 3.3% to 33.3%. This indicates possible reporting bias. How the study might affect research, practice or policy 1. The authors state that...’ There is a critical need for sustainable and affordable diagnostic tests suitable for use in resource limited settings for neonates...’. No data is provided on lack of availability of tests. The study is mixing poor utilization of tests by health care workers with non/ less availability of tests.
--	--

VERSION 1 – AUTHOR RESPONSE

Reviewer: 2. Dr. Vishal Tewari, Command Hospital (SC) and Armed Forces Medical College
Background

1. The study aim mention’s ..’to describe the diagnostic criteria used by clinicians..’. What is the study hypothesis?

Response: In this descriptive study we did not have a study hypothesis.

2. Author states...' However, there is limited information on the impact of guidelines in low-resource settings, where challenges with resources for health are greatest (14)'. How is this related to the current study?

Response: Thank you. We feel that our study provides a further insight into diagnostic practice in low resource settings; however, a further research gap is to improve understanding of how useful guidelines are in such settings (including use of diagnostic technologies) towards improving neonatal health. We have added a comment in the conclusion line 280-281.

3. The diagnosis of neonatal illness in inborn neonates' e.g. preterm respiratory distress syndrome is entirely clinical and hematological and radiological investigations support the diagnosis. In outborn neonates the role of lab investigations to establish and confirm the diagnosis is much more.

Response: Thank you, we agree with this and had added a comment from line 237 to emphasise the importance of laboratory and radiological investigations to complement clinical diagnosis.

4. Patterns of neonatal morbidities in inborn and outborn neonates differ and while it is possible that without adequate lab, radiological, ultrasound and echocardiogram investigations establishing an accurate diagnosis may not be possible in outborn babies, diagnosis in inborn babies and initiation of treatment is almost always clinical.

Response: As per above, we hope our edit emphasises this important point. In our experience we feel that for inborn patients, whilst clinical diagnosis is important, the overlapping clinical features of many neonatal conditions mean that in many cases further investigations are required to confirm diagnosis, as outlined in paragraph beginning line 202.

5. Availability of guidelines for management of common neonatal conditions even though essential for improving neonatal care, what impact does its non-availability in a country in these times when open-source information and guidelines from WHO and similar nations are replete needs to be elaborated.

Response: Thank you for this comment. We agree that further research is needed to understand the impact of guidelines on improving neonatal health (please see additional comment from line 280). However, we feel that guidelines must be designed in relation to the setting and level of care in which they will be used, as the ability of a unit to comply to generic guidelines will vary between different countries and units according to available diagnostic resources and access costs, as outlined in our conclusion. Therefore, context relevant guidelines are likely to be more effective at improving diagnostic practices.

6. What international initiatives for improving diagnosis are being referred to?

Response: We have rephrased this to international efforts, which includes references 5, 15 and 16.

Material and methods

1. Design and setting: The study is a prospective observational study over a 6 month period from 5 tertiary care hospitals and 2 district hospitals from 2 African nations experiencing

high NMR. Since the study is from Urban centers and tertiary care centers, the results of the study may have very little import on development of national policies for reducing NMR. This must be mentioned as a limitation of the study.

Response: Thank you, we have included this in our limitations section.

2. Design and setting: Did the tertiary care neonatal set-ups not have facilities for assisted invasive ventilation?

Response: No, however there was access to bubble CPAP only as states in our methods.

3. More details about the neonatal units enrolling the cases is required. How many bedded? What was the occupancy? Busy neonatal units may not be able to conduct all requisite hematological, biochemistry and radiographic examinations necessary for making a diagnosis.

Response: We have added a new appendix (1) which provides further details about the NNU bed numbers and staffing levels. We hope this supplements the information provided about the units in the methods section. Unfortunately, we do not have any data about bed occupancy during the study period.

4. In what proportion of cases was the diagnosis revised? Can we answer this?

Response: Unfortunately, we did not collect information regarding revision of diagnoses.

5. What was the reason for death and did the cause of death differ from the diagnosis at admission? Can we answer this?

Response: We have already reported cause of death in a previous publication (<https://doi.org/10.1371/journal.pone.0244109>). We have not reported here to avoid repetition and because we do not think this is directly relevant to the challenge of making diagnoses.

6. Was post-mortem examination done in the fatal cases? Was the provisional diagnosis revised following the post-mortem examination?

Response: No post-mortems were performed.

7. Investigations – haematological, biochemistry, acute phase reactants, radiological, ultrasound and echo play a vital role in guiding management but only help to complement the diagnosis in neonatal care algorithms.

Response: Thank you, we hope we have emphasised this point in our addition from line 237. We also hope paragraph beginning line 202 explains the role of laboratory and radiological diagnostics in supporting diagnosis.

8. Clinical staff completed the case reporting forms. Did it include Doctors and Nurses? What steps were taken to ensure accuracy of data entry in the forms?

Response: Case report forms were completed by medical staff and we have clarified this in line 119 by changing “clinical staff” to “clinicians”. Clinicians had varying levels of experience but we have clarified that a senior clinician reviewed all completed forms (line 125).

9. Recording ‘.....a single episode for each condition in each infant...’ this sentence is not clear. Does it mean only one of the multiple diagnosis in a neonate was recoded? This can be a reason for reporting bias.

Response: Thank you, we have added a comment which hopefully clarifies this on line 120-122. Multiple diagnoses in a single neonate were included in the study but only the first instance of each condition.

10. Diagnostic criteria used requires to be reanalyzed with:-
 - a. Distinction between Suspect, Possible and Confirmed sepsis,

Response: Due to limited laboratory support, we did not differentiate as suggested; we recorded suspected sepsis with criteria described from line 118.

- b. Distinction between neonatal encephalopathy of HIE vs. non-HIE origin.

Response: We focussed on birth asphyxia/HIE as the most common neonatal problem and did not include other causes of encephalopathy.

- c. Distinction between Feed Intolerance, NEC and septic ileus,

Response: Because of the similarities/overlap in the clinical presentation between these problems, our approach was to group these together under “abdominal conditions” and then report the parameters (clinical, lab, imaging) that were used.

- d. Distinction between preterm RDS, congenital pneumonia, TTN, MAS etc.

Response: As for c above, we have grouped these under “respiratory conditions”

11. What parameters were used on cranial ultrasound for making the diagnosis of birth asphyxia?

Response: We have not included this as it is not directly relevant to our purpose. Our focus was whether cranial ultrasound was undertaken in these infants (rather than how the findings/interpretation of cranial ultrasound and other investigations were used in diagnosis). We hope this is satisfactory.

12. What about mortality from neonatal lesions like congenital cardiac lesions, PPHN, congenital anomalies which could have any one or more of the clinical syndromes included in the study i.e. sepsis, respiratory distress, birth asphyxia and abdominal distention.

Response: As in 5. above, we have already reported cause of death in a previous publication (<https://doi.org/10.1371/journal.pone.0244109>). We have not reported here to avoid repetition and because we do not think this is directly relevant to the challenge of making diagnoses.

Results

1. With more than 90% births in institutional setting and not home-births in this study, any further reduction in neonatal mortality rate will result from improvement in infrastructure and resources i.e. manpower, training and equipment.

Response: Thank you, we agree and have added a comment from line 278.

2. Overall 16.6% mortality in institutional setting is exceptionally high and typical of resource constrained settings.

Response: Thank you for this comment. We agree this is very high, however as mentioned in our previous paper (<https://doi.org/10.1371/journal.pone.0244109>), this is in keeping with mortality data from other NNUs in sub-Saharan Africa, some examples of which are below:

-Desalew A, Sintayehu Y, Teferi N, Amare F, Geda B, Worku T, Abera K, Asefaw A. Cause and predictors of neonatal mortality among neonates admitted to neonatal intensive care units of public hospitals in eastern Ethiopia: a facility-based prospective follow-up study. *BMC pediatrics*. 2020 Dec;20(1):1-1.

- Tekleab AM, Amaru GM, Tefera YA. Reasons for admission and neonatal outcome in the neonatal care unit of a tertiary care hospital in Addis Ababa: a prospective study. *Research and reports in neonatology*. 2016;6:17.

As mortality was not a key outcome measure in this study we have not expanded on this further to avoid repetition. We hope this is acceptable.

3. There is a remarkable difference in the morbidity patterns handled between the enrolling centers e.g. ELBW rates vary from 3.3% to 33.3%. This indicates possible reporting bias.

Response: Thank you. Rather than reporting bias, we consider that this reflects differences in case mix amongst the newborns admitted to the network NNUs. A strength of our study was engaging NNUs with varied locations/case mix.

How the study might affect research, practice or policy

1. The authors state that....' There is a critical need for sustainable and affordable diagnostic tests suitable for use in resource limited settings for neonates...'. No data is provided on lack of availability of tests. The study is mixing poor utilization of tests by health care workers with non/ less availability of tests.

Response: Thank you for this comment. Whilst we have unfortunately not provided data on lack of availability of tests, in our experience working in these settings and as demonstrated through our referenced material, low use of diagnostic technologies in LMICs is often a complex problem due to multiple factors including them not being present in a facility, and even when they are, they are not serviced or are not working properly, or families do not have the funds to pay for them. We have therefore tried to capture that this is likely a multifactorial issue resulting in overall low use of diagnostic technologies. This section has been removed and we hope that our discussion and conclusion emphasise the many potential factors contributing to our results. We have also amended the sentence beginning line 221 to include the role that lack of utilisation of diagnostics by staff may have.